# MASC: Metal-Aware Sampling and Correction via Reinforcement Learning for Accelerated MRI

**Zhengyi Lu**[1]                                    ZHENGYI.LU@VANDERBILT.EDU

**Ming Lu**[2]                                         MING.LU@VUMC.ORG

**Chongyu Qu**[1]                                   CHONGYU.QU@VANDERBILT.EDU

**Junchao Zhu**[1]                                 JUNCHAO.ZHU@VANDERBILT.EDU

**Junlin Guo**[1]                                    JUNLIN.GUO@VANDERBILT.EDU

**Marilyn Lionts**[1]                             MARILYN.M.LIONTS@VANDERBILT.EDU

**Yanfan Zhu**[1]                                   YANFAN.ZHU@VANDERBILT.EDU

**Yuechen Yang**[1]                               YUECHEN.YANG@VANDERBILT.EDU

**Tianyuan Yao**[1]                               TIANYUAN.YAO@VANDERBILT.EDU

**Jayasai Rajagopal**[3]                         RAJAGOPALJR@ORNL.GOV

**Bennett Allan Landman**[1]               BENNETT.LANDMAN@VANDERBILT.EDU

**Xiao Wang**[3]                                     WANGX2@ORNL.GOV

**Xinqiang Yan**[2]                               XINQIANG.YAN@VUMC.ORG

**Yuankai Huo**[1]                                 YUANKAI.HUO@VANDERBILT.EDU

[1] *Vanderbilt University, Nashville, TN, USA 37215*

[2] *Vanderbilt University Medical Center, Nashville, TN, USA 37232*

[3] *Oak Ridge National Laboratory, Oak Ridge, TN, USA 37831*

**Editors:** Accepted for publication at MIDL 2026

## Abstract

Metal implants in MRI cause severe artifacts that degrade image quality and hinder clinical diagnosis. Traditional approaches address metal artifact reduction (MAR) and accelerated MRI acquisition as separate problems. We propose MASC, a unified reinforcement learning framework that jointly optimizes metal-aware k-space sampling and artifact correction for accelerated MRI. To enable supervised training, we construct a paired MRI dataset using physics-based simulation, generating k-space data and reconstructions for phantoms with and without metal implants. This paired dataset provides simulated 3D MRI scans with and without metal implants, where each metal-corrupted sample has an exactly matched clean reference, enabling direct supervision for both artifact reduction and acquisition policy learning. We formulate active MRI acquisition as a sequential decision-making problem, where an artifact-aware Proximal Policy Optimization (PPO) agent learns to select k-space phase-encoding lines under a limited acquisition budget. The agent operates on undersampled reconstructions processed through a U-Net-based MAR network, learning patterns that maximize reconstruction quality. We further propose an end-to-end training scheme where the acquisition policy learns to select k-space lines that best support artifact removal while the MAR network simultaneously adapts to the resulting undersampling patterns. Experiments demonstrate that MASC's learned policies outperform conventional sampling strategies, and end-to-end training improves performance compared to using a frozen pre-trained MAR network, validating the benefit of joint optimization. Cross-dataset experiments on FastMRI with physics-based artifact simulation further confirm generalization to realistic clinical MRI data. The code and models of MASC have been made publicly available: https://github.com/hrlblab/masc

**Keywords:** MRI reconstruction, k-space sampling, metal artifact reduction, reinforcement learning.

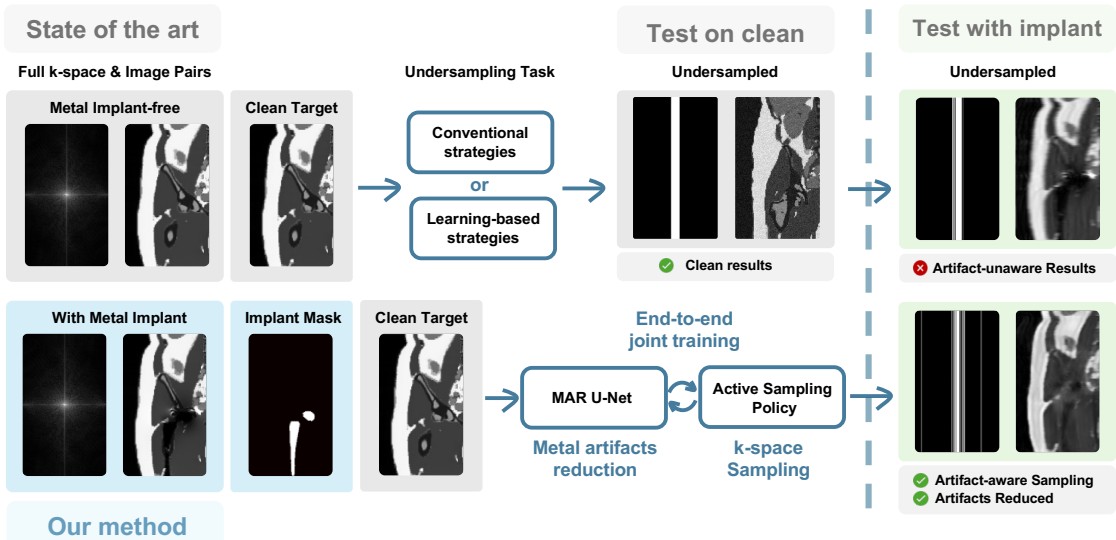

Figure 1: **Overview of MASC.** *Top:* Existing approaches using conventional or learning-based sampling strategies are trained on metal-free data and produce clean results when tested on implant-free cases, but fail when applied to metal-corrupted data, yielding artifact-unaware reconstructions. *Bottom:* MASC jointly optimizes a MAR U-Net and active sampling policy through end-to-end training on metal-corrupted data, enabling artifact-aware k-space sampling and effective artifact reduction.

## 1. Introduction

Magnetic resonance imaging (MRI) is a powerful diagnostic tool that provides excellent soft tissue contrast without ionizing radiation. However, two major challenges limit its clinical utility in certain patient populations: the presence of metal implants, which cause severe image artifacts due to susceptibility differences and signal voids (Hargreaves et al., 2011), and the inherently long scan times required to acquire fully-sampled k-space data (Liang and Lauterbur, 2000; Moratal et al., 2008; Knoll et al., 2020). While both challenges have been extensively studied, existing approaches typically address them in isolation—metal artifact reduction (MAR) methods assume fully-sampled acquisitions, while accelerated MRI techniques focus on artifact-free subjects.

Metal implants are increasingly prevalent in patient populations, with hip replacements, spinal fusion hardware, and dental implants affecting millions of individuals annually (Maradit Kremers et al., 2015; Hegde et al., 2023). These metallic structures induce local magnetic field inhomogeneities that distort the MRI signal, manifesting as signal voids, geometric distortions, and bright streaking artifacts in reconstructed images (Koch et al., 2010; Hargreaves et al., 2011). Such artifacts can obscure adjacent anatomical structures critical for diagnosis, particularly in post-operative imaging where assessment of tissue surrounding the implant is essential. Accelerated MRI acquisition has emerged as a complementary research

direction, aiming to reduce scan times by acquiring only a subset of k-space lines (Wang et al., 2016; Han et al., 2019). Recent work has explored jointly optimizing fixed undersampling patterns with reconstruction networks (Zibetti et al., 2022), while reinforcement learning approaches frame k-space acquisition as a sequential decision-making problem, where an agent learns which phase-encoding lines to acquire to maximize reconstruction quality under a limited sampling budget (Zhang et al., 2019; Pineda et al., 2020; Bakker et al., 2020; Yen et al., 2024).

However, existing learned acquisition methods do not account for metal artifacts, and their interaction with artifact reduction remains unexplored. As illustrated in Figure 1, applying conventional active sampling policies to metal-corrupted data produces reconstructions that retain significant artifacts, since these policies were not designed to handle the unique k-space signatures induced by metal implants.

In this work, we propose MASC (Metal-Aware Sampling and Correction via Reinforcement Learning for Accelerated MRI), a unified framework that jointly addresses metal artifact reduction and accelerated MRI acquisition. Our approach formulates active k-space acquisition as a Markov decision process (Sutton and Barto, 2018), where an artifact-aware Proximal Policy Optimization (PPO) agent (Schulman et al., 2017) learns to select phase-encoding lines while a U-Net-based (Ronneberger et al., 2015) MAR network processes the undersampled reconstructions. To enable supervised training, we construct a paired dataset using the open-source physics-based MRI simulator (Zochowski et al., 2024), which generates realistic TSE k-space data from digital phantoms derived from the AutoPET CT dataset (Gatidis et al., 2022) via multi-tissue segmentation. Virtual metal implants are added to produce exactly matched clean-metal pairs with physically accurate off-resonance artifacts, providing ground truth supervision. Crucially, we introduce an end-to-end training scheme where both components co-adapt: the acquisition policy learns which k-space lines best support artifact removal, while the MAR network adapts to handle the specific undersampling patterns produced by the learned policy (Figure 1, bottom). This co-adaptation addresses a fundamental limitation of frozen MAR approaches—networks trained on fully-sampled data may not optimally handle the aliasing artifacts introduced by aggressive undersampling. We evaluate our approach against conventional sampling strategies and two learning-based RL methods, demonstrating that the learned policies achieve superior reconstruction quality, with end-to-end training providing measurable improvements over frozen MAR baselines. We further validate cross-dataset generalization ability through evaluation on the FastMRI knee dataset (Zbontar et al., 2018; Knoll et al., 2020) with physics-based metal artifact simulation (Kwon et al., 2018), confirming successful transfer to realistic clinical MRI data. Our contributions are:

- To the best of our knowledge, the first study to address accelerated MRI reconstruction in the presence of metal artifacts.

- A physics-based paired hip MRI k-space dataset with simulated metal artifacts enabling supervised training for artifact reduction and reward computation for policy learning.

- An end-to-end artifact-aware reinforcement learning framework for joint optimization of k-space acquisition and metal artifact reduction.

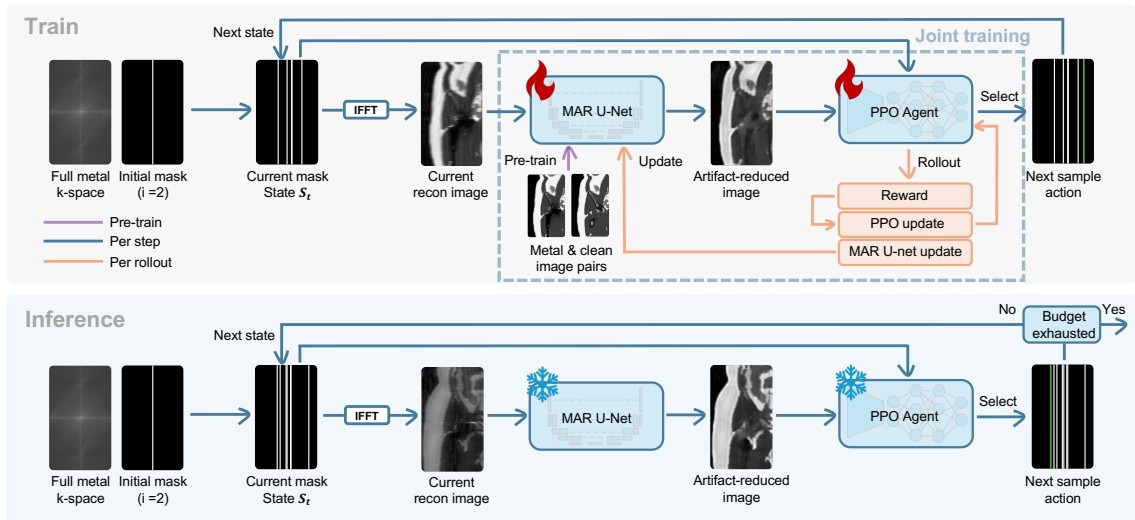

Figure 2: **MASC training and inference pipeline.** *Top (Train):* Starting from full metal-corrupted k-space, the current mask selects acquired lines for IFFT reconstruction. The MAR U-Net processes the reconstruction to produce artifact-reduced images, which the artifact-aware PPO agent observes to determine the next sampling action. Color-coded arrows indicate operation frequency: purple for pre-training (MAR U-Net on paired metal-clean images), blue for per-step operations (sequential k-space acquisition), and orange for per-rollout updates (reward computation, PPO policy update, and MAR U-Net fine-tuning). Fire icons denote trainable networks. *Bottom (Inference):* Both networks are frozen (snowflake icons). The agent iteratively selects k-space lines until the acquisition budget is exhausted.

## 2. Methods

### 2.1. Problem Formulation

We formulate active MRI acquisition as a Markov Decision Process (MDP) (Sutton and Barto, 2018). As shown in Figure 2, the state at time step $t$ is defined as $s_t = (I_t, M_t)$, where $I_t \in \mathbb{R}^{H \times W}$ represents the current magnitude reconstruction obtained from partially sampled k-space, and $M_t \in \{0, 1\}^{N_{\mathrm{pe}}}$ is a binary mask indicating acquired phase-encoding lines. The reconstruction is computed via inverse Fourier transform as $I_t = |\mathcal{F}^{-1}(K \odot M_t)|$, where $K \in \mathbb{C}^{H \times W}$ denotes the full k-space data (Liang and Lauterbur, 2000; Moratal et al., 2008).

The action space consists of selecting a single unacquired phase-encoding line. Upon selecting action $a_t$, the mask is updated as $M_{t+1}[a_t] = 1$, and the agent receives a reward based on reconstruction quality improvement:

$$r_t = \alpha \cdot (Q(I_{t+1}, I^*) - Q(I_t, I^*)) \tag{1}$$

where $I^*$ is the ground truth image, $\alpha$ is a scaling factor, and $Q$ measures reconstruction quality by combining Structural Similarity Index (SSIM) (Wang et al., 2004) and Normalized Mean Squared Error (NMSE):

$$Q(I, I^*) = \lambda_{\text{ssim}} \cdot \text{SSIM}(I, I^*) + \lambda_{\text{nmse}} \cdot (1 - \text{NMSE}(I, I^*)) \qquad (2)$$

### 2.2. MASC Framework

Our framework comprises two components: an artifact-aware PPO-based (Schulman et al., 2017) acquisition policy and a U-Net-based MAR network (Ronneberger et al., 2015).

**Acquisition Policy.** The artifact-aware PPO agent uses a shared convolutional encoder for actor and critic branches, processing the concatenated reconstruction and acquisition mask. The actor outputs action probabilities over all k-space lines, with already-acquired lines masked to zero to ensure valid selection until the budget is exhausted. We optimize using the standard PPO clipped objective with Generalized Advantage Estimation (GAE) (Schulman et al., 2016).

**MAR Network.** The MAR network is a U-Net with four encoder and four decoder stages. Each encoder stage doubles the feature channels (64, doubling at each stage to 1024) using two 3×3 convolutions with batch normalization and ReLU, followed by max pooling. The decoder uses bilinear upsampling with skip connections to preserve spatial details. The network employs residual learning (He et al., 2016) where $g_\psi(I) = I + r_\psi(I)$, allowing it to focus on learning artifact corrections rather than full image reconstruction.

**Co-Adaptive Training.** We propose a two-stage training scheme. In Stage 1, the MAR network is first pre-trained on paired metal-corrupted and clean images using:

$$\mathcal{L}_{\text{pretrain}} = \mathcal{L}_{L1} + \lambda \cdot \mathcal{L}_{SSIM} \qquad (3)$$

where $\mathcal{L}_{L1} = \frac{1}{N} \sum_{i=1}^{N} |x_{pred,i} - x_{gt,i}|$ and $\mathcal{L}_{SSIM} = 1 - \text{SSIM}(x_{pred}, x_{gt})$. This combination is effective for training from scratch: L1 provides robust supervision against outliers while SSIM captures perceptual quality.

In Stage 2, both networks are jointly optimized: the artifact-aware PPO agent observes MAR-processed reconstructions with rewards computed as:

$$r_t = \alpha \cdot (Q(g_\psi(I_{t+1}), I^*) - Q(g_\psi(I_t), I^*)) \qquad (4)$$

where $Q$ is our quality metric and $I^*$ is the ground truth. This enables the policy to learn acquisition patterns complementary to the MAR network. Simultaneously, the MAR network is fine-tuned using MSE loss:

$$\mathcal{L}_{\text{finetune}} = \|g_\psi(I_t) - I^*\|_2^2 \qquad (5)$$

We switch from $(L1 + SSIM)$ to MSE during fine-tuning. MSE provides more stable gradients than L1, which helps stabilize training when combined with noisy policy gradient updates. At this stage, the MAR network is already well-initialized and only needs mild regularization to prevent drift, while the RL reward already captures image quality, making additional perceptual losses unnecessary.

This co-adaptive training creates mutual influence: as the MAR network improves on particular acquisition patterns, the policy learns to favor them; conversely, as the policy

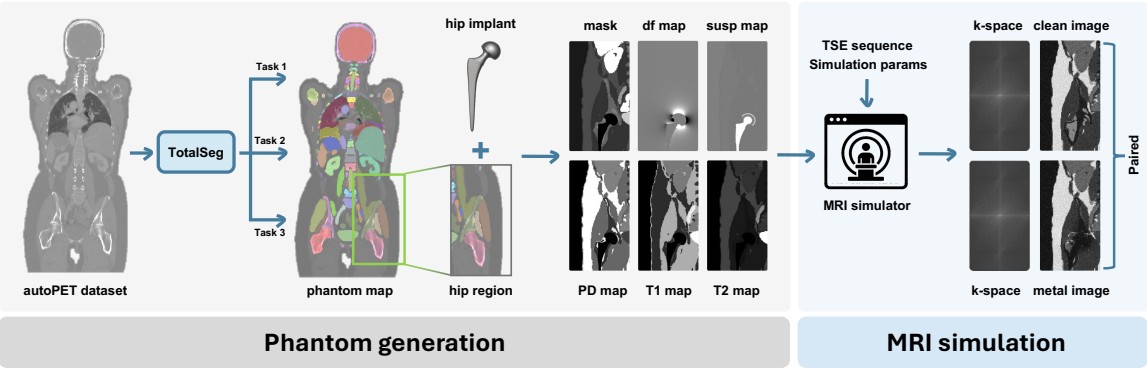

Figure 3: **Dataset construction pipeline.** *Phantom generation:* CT volumes from the autoPET dataset are processed through TotalSegmentator with three complementary tasks to produce multi-tissue phantom maps. The hip region is manually selected and combined with a cobalt-chromium hip implant model to generate tissue property maps including implant mask, off-resonance frequency (df), susceptibility (susp), proton density (PD), T1, and T2. *MRI simulation:* A physics-based simulator with TSE sequence parameters generates paired k-space data and reconstructions: clean images without metal and artifact-corrupted images with the virtual implant, providing exactly matched pairs for supervised training.

converges, the MAR network specializes in those specific undersampling patterns. We alternate updates after each rollout with a lower learning rate for MAR fine-tuning. A detailed pseudo algorithm is provided in Appendix A.

## 3. Data and Experiments

### 3.1. Data

As shown in Figure 3, we construct a paired MRI dataset from 200 subjects in the AutoPET CT/PET dataset (Gatidis et al., 2022) using physics-based metal artifact simulation. This enables supervised training for artifact reduction and reward computation for policy learning. Both phantom segmentations and simulation results were validated by professional radiologists. Detailed subject information is provided in Appendix B.

**Phantom Generation.** For each CT volume, we run TotalSegmentator (Wasserthal et al., 2023) with three complementary tasks: (1) general segmentation for detailed organ delineation, (2) tissue composition for fat and muscle differentiation, and (3) body region for anatomical localization. These segmentations are combined to create multi-tissue 3D phantoms with tissue-specific MRI property maps including proton density, T1, and T2 relaxation times, following established conventions in the field (Segars et al., 2010; Bottomley et al., 1987; Pohmann et al., 2016; Rooney et al., 2007; Bojorquez et al., 2017; Gold et al., 2004; Stanisz et al., 2005). We then manually select the hip region for each subject as the anatomically correct location for hip implant placement.

**MRI Simulation.** We employ the open-source MRI simulator developed by Zochowski et al. (Zochowski et al., 2024) to generate k-space data at 3T field strength. For metal artifact simulation, we use the total hip arthroplasty model provided in the simulator (Edelsbrunner et al., 1983; Mödinger et al., 2023), which comprises a femoral stem, femoral head, acetabular liner, and acetabular cup generated as spherical shells. The implant is modeled with cobalt-chromium (CoCr) alloy with magnetic susceptibility of 900 ppm, while tissue susceptibility values follow the simulator defaults: $-9.05$ ppm for soft tissues and water, $-8.86$ ppm for cortical bone, and $-5.55$ ppm for fat (Schenck, 1996; Smith et al., 2015; Mödinger et al., 2023). We manually fit the implant model into each subject's hip region with position adjustments to ensure anatomically realistic placement. The simulator models susceptibility-induced field inhomogeneities, signal dephasing, and geometric distortions. We simulate a Turbo Spin Echo (TSE) sequence with the following parameters: TR = 4050 ms, TE = 32 ms, readout bandwidth = 710 Hz/pixel, RF bandwidth = 1 kHz and slice thickness = 3 mm (Zochowski et al., 2024).

For each phantom, we generate: (1) clean k-space without metal, (2) metal-corrupted k-space with the virtual implant, and (3) a binary implant mask. Inverse Fourier transform yields paired reconstructions where each metal-corrupted image has an exactly matched clean reference serving as the ground truth target.

**Final Hip MRI Metal Artifacts Dataset.** The resulting dataset contains 200 subjects with 36 slices each, where edge slices are excluded to retain only the region affected by metal artifacts. This paired structure is critical for our framework: the clean reconstructions provide supervision for MAR network training, while the matched pairs enable reward computation during policy learning.

**FastMRI-based Metal Artifacts Dataset.** To enable cross-dataset generalization evaluation, we construct an additional test set using the FastMRI knee dataset (Zbontar et al., 2018; Knoll et al., 2020), which contains realistic clinical MRI scans from NYU Langone Health. Metal artifacts are synthesized following the physics-based simulation method of Kwon et al. (Kwon et al., 2018), modeling off-resonance field distortion of a simplified hip implant (Shi et al., 2017) different from the training TSE hip dataset, RF profile modulation (FWHM = 2.25 kHz), and geometric pile-up/void artifacts from Jacobian-based intensity modulation. Random implant rotation ($\pm 45°$) and translation ($\pm 80$ pixels) are applied to create diverse artifact patterns. This dataset provides an independent test distribution for evaluating generalization beyond our primary training data.

### 3.2. Experimental Setup

We partition the 200 subjects into training (160 subjects), validation (20 subjects), and test (20 subjects) sets, ensuring no subject overlap between splits. Each subject contains 36 slices, yielding 5,760 training, 720 validation, and 720 test slices. All experiments are conducted on an NVIDIA RTX 5090 GPU with 32 GB memory.

**Training Configuration.** The MAR network is pretrained for 100 epochs on paired metal-corrupted and clean images using the loss defined in Equation 3. For end-to-end training, we use Adam optimizer (Kingma and Ba, 2015) with learning rate $3 \times 10^{-4}$ for the artifact-aware PPO agent and $1 \times 10^{-5}$ for the MAR network. PPO hyperparameters include: rollout length of 512 steps, 4 optimization epochs per update, clip range of 0.2,

and entropy coefficient of 0.01. The reward scaling factor is $\alpha = 100$, and the quality metric weights are $\lambda_{\text{ssim}} = 0.5$ and $\lambda_{\text{nmse}} = 0.5$. All experiments use an acceleration factor of $10\times$ (2 initial plus 18 budget acquired lines). Additionally, experiments with a different acceleration factor of $5\times$ (8 initial plus 32 budget acquired lines) are shown in Appendix C.

**Baseline Methods.** We compare against four conventional acquisition strategies and two learned baselines: (1) Center-out, acquiring lines from k-space center outward; (2) Random, uniform random selection; (3) Random-LowBias, random selection biased toward central k-space; (4) Equispaced, fixed equidistant sampling; and (5) Deep Q-Network (DQN) (Mnih et al., 2015); (6) Subject-Specific Double Deep Q-Network (SS-DDQN), a learned policy based on Double DQN for active MRI acquisition (Pineda et al., 2020). Conventional baselines are evaluated both with and without MAR post-processing to enable comparison against both naive sampling and two-stage acquisition-then-reconstruction approaches.

**Ablation Study.** We conduct ablation experiments with five configurations: (1) Clean-trained PPO without MAR; (2) Metal-trained PPO without MAR; (3) Clean PPO + Pre-trained MAR (frozen); (4) Metal PPO + Pretrained MAR (frozen); and (5) MASC with joint co-adaptive optimization. The comparison between (4) and (5) isolates the benefit of end-to-end training versus applying MAR only at inference.

### 3.3. Evaluation Metrics

We assess reconstruction quality using SSIM, Peak Signal-to-Noise Ratio (PSNR), Mean Squared Error (MSE), NMSE, and Mean Absolute Error (MAE). All metrics are computed on final reconstructions after the acquisition budget is exhausted, reported as mean $\pm$ standard deviation across the test set.

## 4. Results

### 4.1. Comparison with Baseline Acquisition Strategies

Table 1 presents the quantitative comparison between our proposed MASC method and baseline acquisition strategies on the test set. Baseline methods are evaluated both with and without MAR post-processing to compare against naive sampling strategies and two-stage acquisition-then-reconstruction pipelines.

Among conventional baselines without MAR, Center-out achieves the best performance, consistent with the understanding that low-frequency k-space components contain the majority of image energy and structural information. Random-LowBias outperforms uniform Random sampling by biasing selection toward the k-space center, while Equispaced sampling performs worst as it is designed for parallel imaging with coil sensitivity information unavailable in our single-coil setting. DQN and SS-DDQN, learned acquisition policies based on value-based RL, outperform Center-out in MSE and NMSE but underperform in SSIM, suggesting that value-based RL can learn useful sampling patterns but may not fully capture perceptual quality objectives.

When MAR post-processing is applied to baseline acquisitions, reconstruction quality improves substantially across all methods. Interestingly, Random and Random-LowBias with MAR achieve strong performance (SSIM > 0.69), as incoherent sampling patterns

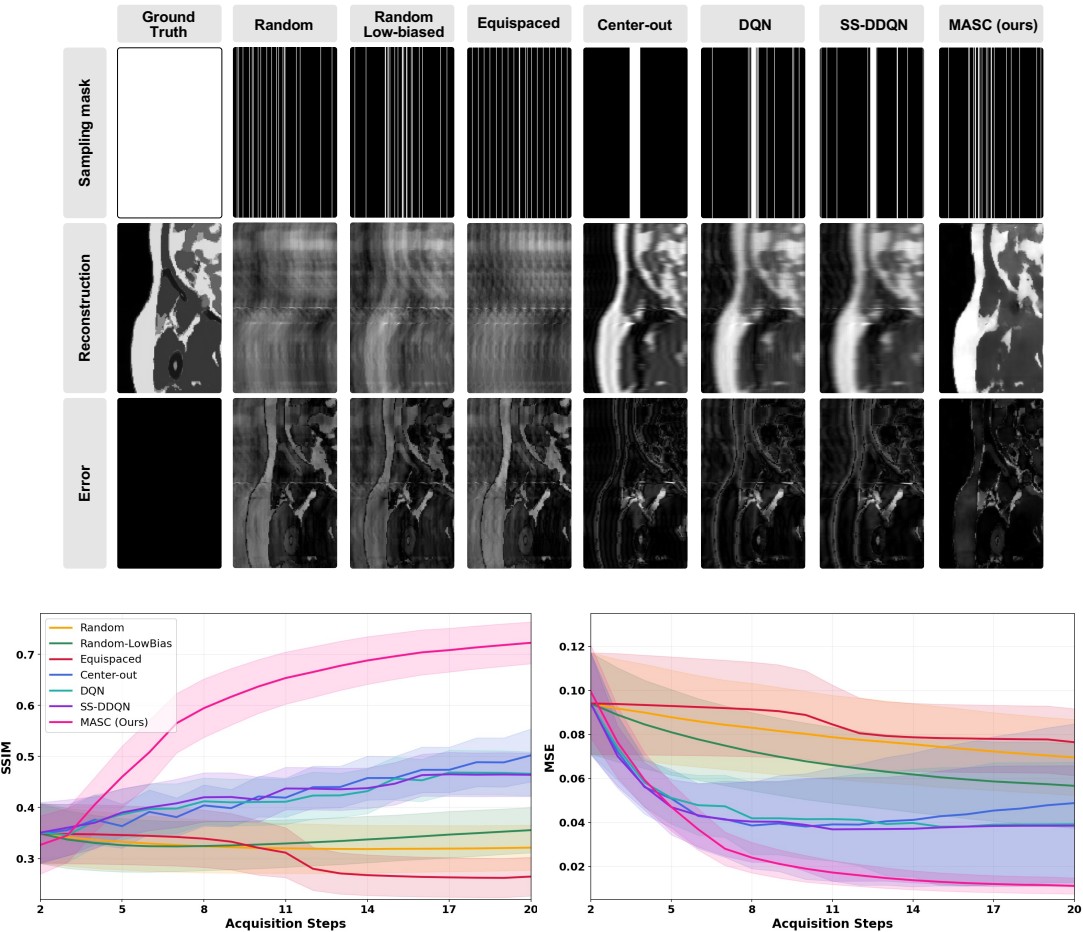

Figure 4: **Comparison of acquisition strategies.** *Top:* Ground truth, sampling masks, reconstructions, and error maps for each method including four conventional baselines (Random, Random Low-biased, Equispaced, Center-out), learned baseline (DQN, SS-DDQN), and our MASC. MASC produces reconstructions closest to ground truth with substantially darker error maps. *Bottom:* SSIM and MSE versus number of acquired k-space lines. Shaded regions indicate the std. MASC demonstrates superior performance throughout acquisition, with the gap widening as more lines are acquired.

provide diverse k-space coverage that benefits artifact correction. In contrast, Equispaced with MAR remains the weakest (SSIM = 0.41), indicating that structured aliasing patterns are difficult to correct even with dedicated MAR networks. DQN and SS-DDQN with MAR achieve moderate performance (SSIM of 0.64 and 0.62, respectively), falling between the random and deterministic strategies.

Our MASC method substantially outperforms all baselines across every metric, including those enhanced with MAR post-processing. Compared to the best two-stage baseline

Table 1: **Comparison with baseline acquisition strategies at** $10\times$ **acceleration (2 initial + 18 budget lines).** Baselines are evaluated with and without MAR. Results are mean $\pm$ std on the test set. Best results in **bold**. All improvements are statistically significant ($p < 0.001$, paired t-test).

| Method | MAR | SSIM $\uparrow$ | PSNR $\uparrow$ | MSE $\downarrow$ | NMSE $\downarrow$ | MAE $\downarrow$ |
|---|---|---|---|---|---|---|
| Random | $\times$ | $0.3214 \pm 0.0440$ | $11.72 \pm 1.11$ | $0.0695 \pm 0.0173$ | $0.3374 \pm 0.0929$ | $0.2199 \pm 0.0333$ |
| | $\checkmark$ | $0.6949 \pm 0.0463$ | $18.79 \pm 1.42$ | $0.0140 \pm 0.0049$ | $0.0680 \pm 0.0243$ | $0.0708 \pm 0.0143$ |
| Random-LowBias | $\times$ | $0.3556 \pm 0.0444$ | $12.70 \pm 1.38$ | $0.0566 \pm 0.0193$ | $0.2696 \pm 0.0718$ | $0.1877 \pm 0.0332$ |
| | $\checkmark$ | $0.6929 \pm 0.0516$ | $19.10 \pm 1.45$ | $0.0131 \pm 0.0056$ | $0.0635 \pm 0.0241$ | $0.0716 \pm 0.0170$ |
| Equispaced | $\times$ | $0.2648 \pm 0.0380$ | $11.26 \pm 0.92$ | $0.0764 \pm 0.0152$ | $0.3716 \pm 0.0846$ | $0.2331 \pm 0.0297$ |
| | $\checkmark$ | $0.4085 \pm 0.0478$ | $12.82 \pm 1.05$ | $0.0538 \pm 0.0131$ | $0.2607 \pm 0.0659$ | $0.1546 \pm 0.0237$ |
| Center-out | $\times$ | $0.5022 \pm 0.0519$ | $14.23 \pm 3.17$ | $0.0488 \pm 0.0360$ | $0.2249 \pm 0.1352$ | $0.1507 \pm 0.0619$ |
| | $\checkmark$ | $0.5532 \pm 0.0568$ | $16.33 \pm 1.47$ | $0.0248 \pm 0.0101$ | $0.1194 \pm 0.0421$ | $0.1107 \pm 0.0252$ |
| DQN | $\times$ | $0.4660 \pm 0.0429$ | $14.88 \pm 2.53$ | $0.0392 \pm 0.0282$ | $0.1823 \pm 0.1049$ | $0.1362 \pm 0.0475$ |
| | $\checkmark$ | $0.6411 \pm 0.0601$ | $18.18 \pm 1.49$ | $0.0163 \pm 0.0077$ | $0.0791 \pm 0.0334$ | $0.0846 \pm 0.0219$ |
| SS-DDQN | $\times$ | $0.4637 \pm 0.0425$ | $14.90 \pm 2.41$ | $0.0385 \pm 0.0273$ | $0.1798 \pm 0.1017$ | $0.1355 \pm 0.0460$ |
| | $\checkmark$ | $0.6248 \pm 0.0641$ | $17.78 \pm 1.46$ | $0.0178 \pm 0.0074$ | $0.0865 \pm 0.0350$ | $0.0868 \pm 0.0207$ |
| MASC (Ours) | $\checkmark$ | $\mathbf{0.7224 \pm 0.0408}$ | $\mathbf{19.73 \pm 1.28}$ | $\mathbf{0.0111 \pm 0.0036}$ | $\mathbf{0.0546 \pm 0.0190}$ | $\mathbf{0.0640 \pm 0.0121}$ |

(Random + MAR), MASC achieves 4.0% improvement in SSIM and 20.7% reduction in MSE. Compared to Center-out without MAR, the improvements are 43.9% in SSIM and 77.3% reduction in MSE. This improvement stems from end-to-end joint optimization, where MASC learns sampling patterns tailored to the MAR network's correction capabilities.

Figure 4 provides qualitative comparison and acquisition dynamics. The top panel shows that Random and Random-LowBias produce severe aliasing, while Equispaced results in structured aliasing patterns. Center-out preserves gross anatomy but loses fine details. DQN and SS-DDQN show improved structure but retain visible artifacts. In contrast, MASC produces reconstructions closest to ground truth with substantially darker error maps.

The bottom panel reveals the temporal dynamics of acquisition. MASC exhibits a steeper improvement trajectory in early steps, suggesting the learned policy prioritizes k-space lines with maximal information gain. The widening performance gap indicates that MASC's sequential decisions compound beneficially, whereas baseline methods lack this adaptive capability. MASC also maintains consistently lower variance (narrower shaded regions), demonstrating robust performance across diverse anatomical structures and artifact patterns.

## 4.2. Ablation Study

We conduct an ablation study to analyze the contribution of each component in our proposed MASC framework. Table 2 presents results of five configurations that systematically evaluate the effects of training data, MAR integration, and end-to-end optimization. Further visualizations are shown in Appendix E.

Table 2: **Ablation study on training strategy components at $10\times$ acceleration.** Configurations progressively add metal-aware training, MAR integration, and end-to-end optimization. Results are mean $\pm$ std on the test set. Best results in **bold**. MASC significantly outperforms all ablation variants ($p < 0.001$, paired t-test).

| Configuration | SSIM ↑ | PSNR ↑ | MSE ↓ | NMSE ↓ | MAE ↓ |
|---|---|---|---|---|---|
| Clean-trained PPO | $0.5120 \pm 0.0559$ | $14.47 \pm 3.19$ | $0.0463 \pm 0.0348$ | $0.2137 \pm 0.1309$ | $0.1465 \pm 0.0606$ |
| Metal-trained PPO | $0.4758 \pm 0.0465$ | $14.99 \pm 2.47$ | $0.0373 \pm 0.0220$ | $0.1766 \pm 0.0926$ | $0.1341 \pm 0.0425$ |
| Clean PPO + Pretrained MAR | $0.6640 \pm 0.0600$ | $17.08 \pm 1.68$ | $0.0212 \pm 0.0095$ | $0.1020 \pm 0.0400$ | $0.0945 \pm 0.0262$ |
| Metal PPO + Pretrained MAR | $0.6874 \pm 0.0621$ | $17.72 \pm 1.59$ | $0.0182 \pm 0.0077$ | $0.0888 \pm 0.0384$ | $0.0820 \pm 0.0213$ |
| MASC (Ours) | $\mathbf{0.7224 \pm 0.0408}$ | $\mathbf{19.73 \pm 1.28}$ | $\mathbf{0.0111 \pm 0.0036}$ | $\mathbf{0.0546 \pm 0.0190}$ | $\mathbf{0.0640 \pm 0.0121}$ |

Comparing Clean-trained PPO and Metal-trained PPO reveals how metal artifacts affect policy learning without MAR post-processing. Clean-trained PPO achieves higher SSIM while Metal-trained PPO yields lower MSE, suggesting a trade-off: artifact-free training preserves structural learning, but metal-aware training enables partial adaptation to the artifact distribution. Metal artifacts corrupt the reconstruction used for reward computation, effectively adding noise to the policy gradient.

Adding pretrained MAR substantially improves all configurations, with Metal PPO + Pretrained MAR slightly outperforming its clean-trained counterpart. This suggests that when MAR corrects artifacts at inference, the metal-trained policy's implicit knowledge of artifact locations becomes beneficial, learning to sample k-space regions that provide complementary information for correction.

Our proposed MASC framework achieves the best performance across all metrics, improving SSIM by 5.1% and reducing MSE by 39.0% compared to Metal PPO + Pretrained MAR. This improvement stems from resolving the train-test distribution mismatch: in post-hoc approaches, the MAR network is trained on fully-sampled data but applied to undersampled reconstructions with unfamiliar aliasing artifacts. End-to-end training exposes the MAR network to actual undersampling patterns while allowing the policy to discover acquisition strategies that maximize correction capability. The consistently lower standard deviation further indicates that co-adaptation produces more robust reconstructions.

### 4.3. Cross-Dataset Generalization

We evaluate generalization by testing our model (trained on TSE phantom data) on the FastMRI-based dataset, which uses a different hip implant geometry than the training set. Table 3 presents the quantitative results, where all baselines are post-processed with the same MAR network for fair comparison.

MASC maintains superior performance on the unseen dataset, achieving 3.0% improvement in SSIM and 23.1% reduction in MSE compared to the best baseline, despite generalizing across both data distribution shifts and unseen implant geometry. The learned value-based policies (DQN and SS-DDQN) exhibit degraded generalization compared to simple random sampling, suggesting overfitting to training-specific artifact patterns. In contrast, MASC demonstrates robust transfer, indicating that joint optimization learns more generalizable representations than acquisition-only learning.

Table 3: **Cross-dataset generalization at** $10\times$ **acceleration.** All methods are trained on our physics-based simulation dataset and evaluated on the FastMRI-based Metal Artifacts Dataset. The baselines are post-processed with MAR for fair comparison. Results are mean $\pm$ std on the test set. Best results in **bold**. MASC significantly outperforms all baselines ($p < 0.001$, paired t-test).

| Method (with MAR) | SSIM ↑ | PSNR ↑ | MSE ↓ | NMSE ↓ | MAE ↓ |
|---|---|---|---|---|---|
| Random | $0.5562 \pm 0.0434$ | $20.41 \pm 2.73$ | $0.0111 \pm 0.0079$ | $0.0642 \pm 0.0762$ | $0.0773 \pm 0.0282$ |
| Random-LowBias | $0.5636 \pm 0.0428$ | $20.41 \pm 2.56$ | $0.0108 \pm 0.0064$ | $0.0558 \pm 0.0305$ | $0.0767 \pm 0.0245$ |
| Equispaced | $0.5402 \pm 0.0465$ | $20.12 \pm 2.57$ | $0.0116 \pm 0.0071$ | $0.0663 \pm 0.0652$ | $0.0790 \pm 0.0262$ |
| Center-out | $0.5072 \pm 0.0552$ | $17.79 \pm 2.67$ | $0.0199 \pm 0.0119$ | $0.0994 \pm 0.0417$ | $0.1096 \pm 0.0373$ |
| DQN | $0.5376 \pm 0.0483$ | $19.52 \pm 2.36$ | $0.0129 \pm 0.0068$ | $0.0686 \pm 0.0375$ | $0.0851 \pm 0.0255$ |
| SS-DDQN | $0.5305 \pm 0.0487$ | $19.55 \pm 2.36$ | $0.0127 \pm 0.0066$ | $0.0682 \pm 0.0389$ | $0.0842 \pm 0.0244$ |
| MASC (Ours) | $\mathbf{0.5803 \pm 0.0341}$ | $\mathbf{21.41 \pm 2.25}$ | $\mathbf{0.0083 \pm 0.0052}$ | $\mathbf{0.0497 \pm 0.0548}$ | $\mathbf{0.0682 \pm 0.0217}$ |

Interestingly, Center-out shows the worst generalization despite performing competitively on the training dataset, suggesting that its low-frequency bias does not transfer well across different data distributions. MASC also exhibits the lowest variance across all metrics, demonstrating consistent performance across diverse artifact patterns.

## 5. Conclusion

We presented MASC, a unified reinforcement learning framework that jointly optimizes metal-aware k-space sampling and artifact correction for accelerated MRI. Our approach contributes a physics-based paired MRI dataset with simulated metal artifacts from 200 subjects, enabling supervised training for artifact reduction and reward computation for policy learning, along with an end-to-end co-adaptive training scheme where the PPO-based acquisition policy and U-Net-based MAR network mutually adapt to each other. Experiments demonstrate that MASC substantially outperforms both conventional and learned acquisition strategies, as well as two-stage pipelines that apply MAR as post-processing. Compared to the best two-stage baseline, MASC achieves 4.0% improvement in SSIM and 20.7% reduction in MSE; compared to the best conventional baseline without MAR, the improvements are 43.9% in SSIM and 77.3% reduction in MSE at $10\times$ acceleration. Cross-dataset evaluation on the FastMRI-based Metal Artifacts Dataset, which uses a different hip implant geometry than training, demonstrates that MASC generalizes effectively across both data distributions and implant variations. These results demonstrate that joint optimization outperforms sequential approaches where acquisition and reconstruction are treated independently. Ablation studies confirm that end-to-end training provides measurable benefits over using a frozen pre-trained MAR network, validating the importance of joint optimization.

The current limitation of our work is the focus on hip implants at one anatomical region. Future work will extend MASC to diverse implant materials with varying magnetic susceptibilities, different anatomical regions such as spine and knee, and multi-coil settings where parallel imaging can be combined with learned acquisition policies.

## Acknowledgments

This research was supported by NIH R01DK135597 (Huo), DoD HT9425-23-1-0003 (HCY), NSF 2434229 (Huo) and KPMP Glue Grant. This work was also supported by Vanderbilt Seed Success Grant and Vanderbilt-Liverpool Seed Grant. This research was also supported by NIH grants R01EB033385, R01DK132338. We extend gratitude to NVIDIA for their support by means of the NVIDIA hardware grant. This research was also supported by NIH R01 EB031078 (Yan), R21 EB029639 (Yan), R03 EB034366 (Yan). This manuscript has been co-authored by ORNL, operated by UT-Battelle, LLC under Contract No. DE-AC05-00OR22725 with the U.S.Department of Energy.

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

## Appendix A. Additional Implementation Details

Algorithm 1 outlines our co-adaptive training procedure, where the PPO policy and MAR-UNet are jointly optimized after each rollout.

**Algorithm 1: MASC: Joint PPO and MAR-UNet Training**

**Input** : Dataset $\mathcal{D}$, rollout length $N$, scaling factor $\alpha$
**Output:** Trained policy $\pi_\theta$ and MAR-UNet $g_\psi$
Initialize policy $\pi_\theta$, value network $V_\phi$, pretrained MAR-UNet $g_\psi$
**while** *not converged* **do**
    Initialize rollout buffer $\mathcal{B} \leftarrow \emptyset$
    **foreach** *step t in rollout* **do**
        Reset environment if episode done
        Sample action $a_t \sim \pi_\theta(\cdot|s_t)$
        Update mask $M_{t+1} \leftarrow M_t \cup \{a_t\}$
        Reconstruct $I_{t+1} \leftarrow \mathcal{F}^{-1}(k_{\text{full}} \odot M_{t+1})$
        Apply MAR-UNet to obtain $g_\psi(I_{t+1})$
        Compute reward $r_t \leftarrow \alpha \cdot (Q(g_\psi(I_{t+1}), I^*) - Q(g_\psi(I_t), I^*))$
        Store transition $(s_t, a_t, r_t, s_{t+1})$ in $\mathcal{B}$
    **end**
    Update PPO policy $\theta$ and value $\phi$ using $\mathcal{B}$
    Update MAR-UNet $\psi$ with $\mathcal{L}_{\text{finetune}} = \|g_\psi(I_t) - I^*\|_2^2$
**end**
**return** $\pi_\theta$, $g_\psi$

## Appendix B. Additional Dataset Details

We selected 200 subjects from the AutoPET FDG-PET/CT dataset (Gatidis et al., 2022) for MRI simulation. Table 4 summarizes patient demographics.

Table 4: **Dataset demographics (n=200).** Summary of patient characteristics from the AutoPET dataset used for MRI simulation.

| Characteristic | Value |
|---|---|
| Subjects | 200 |
| Age (years) | $62.0 \pm 15.4$ (range: 11–95) |
| **Sex** | |
|    Male | 115 (57.5%) |
|    Female | 84 (42.0%) |
|    Not reported | 1 (0.5%) |
| **Diagnosis** | |
|    Melanoma | 70 (35.0%) |
|    Lung cancer | 65 (32.5%) |
|    Lymphoma | 61 (30.5%) |
|    Negative | 4 (2.0%) |

Table 5: **Comparison with baseline acquisition strategies at $5\times$ acceleration (8 initial $+$ 32 budget lines).** Baselines are evaluated with and without metal artifact reduction (MAR). Results are mean $\pm$ std on the test set. Best results in **bold**. MASC significantly outperforms all baselines ($p < 0.001$, paired t-test).

| Method | MAR | SSIM ↑ | PSNR ↑ | MSE ↓ | NMSE ↓ | MAE ↓ |
|---|---|---|---|---|---|---|
| Random | $\times$ | $0.4037 \pm 0.0351$ | $13.95 \pm 2.37$ | $0.0471 \pm 0.0290$ | $0.2193 \pm 0.1143$ | $0.1518 \pm 0.0453$ |
| | $\checkmark$ | $0.6789 \pm 0.0683$ | $18.36 \pm 1.87$ | $0.0162 \pm 0.0089$ | $0.0760 \pm 0.0358$ | $0.0861 \pm 0.0262$ |
| Random-LowBias | $\times$ | $0.4317 \pm 0.0441$ | $13.59 \pm 2.94$ | $0.0544 \pm 0.0361$ | $0.2525 \pm 0.1441$ | $0.1616 \pm 0.0562$ |
| | $\checkmark$ | $0.6580 \pm 0.0875$ | $18.39 \pm 2.39$ | $0.0175 \pm 0.0156$ | $0.0809 \pm 0.0579$ | $0.0885 \pm 0.0371$ |
| Equispaced | $\times$ | $0.3670 \pm 0.0252$ | $13.77 \pm 2.26$ | $0.0486 \pm 0.0288$ | $0.2254 \pm 0.1120$ | $0.1555 \pm 0.0443$ |
| | $\checkmark$ | $0.5716 \pm 0.0585$ | $17.01 \pm 1.56$ | $0.0214 \pm 0.0099$ | $0.1018 \pm 0.0391$ | $0.0980 \pm 0.0256$ |
| Center-out | $\times$ | $0.5624 \pm 0.1164$ | $12.69 \pm 3.65$ | $0.0709 \pm 0.0446$ | $0.3296 \pm 0.1810$ | $0.1845 \pm 0.0712$ |
| | $\checkmark$ | $0.5599 \pm 0.1114$ | $15.64 \pm 2.71$ | $0.0334 \pm 0.0242$ | $0.1549 \pm 0.0953$ | $0.1313 \pm 0.0504$ |
| DQN | $\times$ | $0.5664 \pm 0.1057$ | $13.06 \pm 3.74$ | $0.0663 \pm 0.0439$ | $0.3074 \pm 0.1773$ | $0.1767 \pm 0.0714$ |
| | $\checkmark$ | $0.5831 \pm 0.1130$ | $16.21 \pm 2.70$ | $0.0296 \pm 0.0237$ | $0.1364 \pm 0.0901$ | $0.1225 \pm 0.0492$ |
| SS-DDQN | $\times$ | $0.5299 \pm 0.0758$ | $13.40 \pm 3.50$ | $0.0605 \pm 0.0418$ | $0.2795 \pm 0.1674$ | $0.1682 \pm 0.0676$ |
| | $\checkmark$ | $0.6145 \pm 0.1095$ | $16.94 \pm 2.57$ | $0.0248 \pm 0.0205$ | $0.1143 \pm 0.0784$ | $0.1103 \pm 0.0438$ |
| MASC (Ours) | $\checkmark$ | $\mathbf{0.7056 \pm 0.0528}$ | $\mathbf{19.41 \pm 1.90}$ | $\mathbf{0.0128 \pm 0.0074}$ | $\mathbf{0.0615 \pm 0.0325}$ | $\mathbf{0.0742 \pm 0.0234}$ |

## Appendix C. Experiments and Results at $5\times$ Acceleration

Table 5 presents results at $5\times$ acceleration. MASC achieves the best performance across all metrics, with 3.9% improvement in SSIM and 21.0% reduction in MSE compared to the best two-stage baseline (Random + MAR). The trends are consistent with $10\times$ acceleration: random sampling strategies benefit most from MAR post-processing, while Center-out shows minimal improvement despite its strong performance without MAR. Value-based RL methods (DQN and SS-DDQN) again fall between random and deterministic strategies when combined with MAR. Notably, the performance gap between MASC and baselines narrows at lower acceleration factors, as the increased k-space budget reduces the importance of optimal line selection.

## Appendix D. Influence of Simulation Parameters

Our simulation uses clinically realistic TSE parameters including readout bandwidth of 710 Hz/pixel, RF bandwidth of 1 kHz, and field strength of 3T. These parameters influence the severity of metal artifacts and thus the task difficulty. Higher readout bandwidth generally reduces susceptibility-induced distortion but lowers SNR, creating a practical trade-off. Field strength also plays a key role since higher field strengths like 7T would produce more severe artifacts due to increased susceptibility effects, while lower field strengths like 0.55T would result in milder artifacts. The implant material matters as well; our cobalt-chromium implant (900 ppm susceptibility) creates substantial artifacts, whereas titanium implants ($\sim$180 ppm) would produce milder distortions. Our parameter choices represent a typical and challenging clinical scenario for hip imaging at 3T, providing a reasonable testbed for evaluating acquisition strategies under metal artifact conditions.

## Appendix E. Additional Ablation Study Visualization

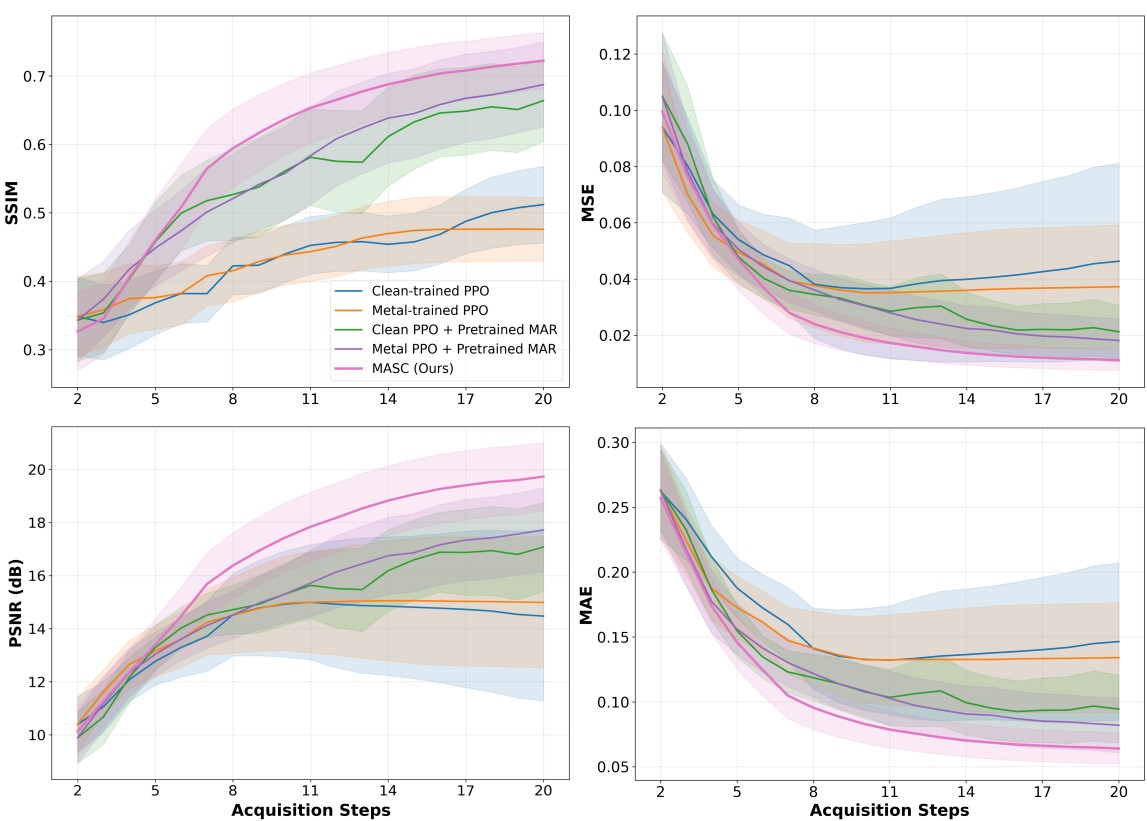

Figure 5: **Ablation study on training strategy components at** $10\times$ **acceleration**. Configurations progressively add metal-aware training, MAR integration, and end-to-end optimization. SSIM, MSE, PSNR and MAE versus number of acquired k-space lines. Shaded regions indicate the std. MASC demonstrates superior performance throughout acquisition, with the gap widening as more lines are acquired.

