# OpenReview forum: "MASC: Metal-Aware Sampling and Correction via Reinforcement Learning for Accelerated MRI"
_MIDL.io/2026/Conference — MIDL 2026 Poster_

### Official Review · Reviewer_rFC2 · 2026-01-04

**Confidence:** 3
**Preliminary Rating:** 4
**Final Rating:** 4

**Summary:**

Metal implants create severe artifacts that disrupt standard accelerated MRI acquisition and reconstruction pipelines. MASC introduces a unified framework that uses reinforcement learning to jointly optimize k-space sampling patterns and a downstream metal artifact reduction network. To enable supervised learning for this task, a physics-based simulation pipeline generates paired k-space data from CT volumes, providing ground-truth "clean" references for metal-corrupted samples. Experiments demonstrate that this co-adaptive approach allows the sampling policy to target k-space lines that maximize the artifact reduction network's performance, outperforming conventional sampling strategies.

**Strengths:**

- Tackling the intersection of active acquisition and metal artifact reduction addresses a specific but highly clinically relevant gap in the current literature.
- Jointly optimizing the PPO agent with the reconstruction network allows the sampling policy to learn non-intuitive patterns specifically designed to mitigate metal-induced spectral corruption.
- Constructing a paired dataset via rigorous physics-based simulation on AutoPET data offers a clever and necessary solution to the impossibility of obtaining real-world ground truth for metal artifacts.
- Ablation studies clearly isolate the benefits of the end-to-end training strategy, proving that the policy effectively adapts to the specific capabilities of the MAR network.
- Visualizations of the acquisition trajectory show the agent learning to prioritize high-information frequency bands, aligning well with MR physics intuition.

**Weaknesses:**

- Validating exclusively on simulated data creates a significant domain gap, as real-world metal artifacts involve complex B0 inhomogeneities and pile-up effects that may not be fully captured by the simulator.
- Limiting the study to single-coil simulations reduces immediate clinical impact, given that parallel imaging is the standard for accelerated MRI in modern scanners.
- Focusing solely on hip implants restricts the evaluation of the policy's robustness, leaving it unclear how the method handles more complex geometries like spinal fixation or dental hardware.
- Comparing primarily against static sampling baselines (Random, Center-out) overlooks stronger recent deep learning-based reconstruction baselines (e.g., VarNet or unrolled networks) that might handle undersampling artifacts better even without active selection.

**Detailed Comments:**

- Discussing the inference latency of the PPO agent is crucial, as active acquisition requires real-time decision-making during the TR intervals of the scanner.
- Expanding the discussion on how the specific simulation parameters (e.g., readout bandwidth) influence the difficulty of the task would provide better context for the results.
- Clarifying whether the reward function's dependence on SSIM and NMSE drives the agent to prioritize high-frequency edges or low-frequency contrast would help interpret the learned masks.
- Providing more details on the "invalid actions masked" step in the policy network would clarify how the agent handles the budget constraints during the rollout.

**Justification Of Final Rating:**

The rebuttal effectively addressed key concerns regarding generalization and fair comparison. The added FastMRI cross-dataset evaluation and DQN baseline comparisons substantially strengthen validation, while clarifications on architecture and losses improve clarity. Despite persisting single-coil limitations and lack of physical phantom validation, the joint optimization framework demonstrates clear methodological value for metal-aware MRI acquisition.

**Justification Of The Preliminary Rating:**

MASC proposes a novel and methodologically sound approach to a complex problem that is often treated in isolation. Creating a physics-based paired dataset to enable reinforcement learning in this domain is a valuable contribution. However, the reliance on single-coil simulations and the lack of real-world validation data limit the immediate clinical applicability and prevent a higher score. Despite these limitations, the joint optimization strategy is promising and relevant to the MIDL community.

**Questions To Address In The Rebuttal:**

- How does the framework perform when tested on implant materials or geometries significantly different from the training set (e.g., Titanium vs. CoCr)?
- Is there a straightforward pathway to extend this method to multi-coil parallel imaging, and does the state space explosion become a bottleneck?
- Have you attempted to apply the trained policy to any real phantom data (even without ground truth) to qualitatively assess if the simulation-learned features transfer to real k-space?

---

> ### Author Response · Authors · 2026-01-25
> **Response to Reviewer rFC2 (1/2)**
>
> We would like to thank you for recognizing the strengths of our paper, including that "addresses a specific but highly clinically relevant gap in the current literature", "aligning well with MR physics intuition", and "offers a clever and necessary solution to the impossibility of obtaining real-world ground truth for metal artifacts". In the following, we provide point-by-point responses to all your concerns.
>
> ---
>
> >W1: *Validating exclusively on simulated data creates a significant domain gap, as real-world metal artifacts involve complex B0 inhomogeneities and pile-up effects that may not be fully captured by the simulator.*
>
> We thank the reviewer for this concern. As the reviewer noted, paired real data is impossible to obtain for metal artifacts. To test generalization, we added **cross-dataset evaluation on FastMRI** using real clinical MRI scans with a different artifact simulation method. MASC still works well on this unseen data (**Table 3**), suggesting our method is not overfitting to one specific simulator. Testing on real implant cases is an interesting future direction.
>
> ---
>
> >W2: *Limiting the study to single-coil simulations reduces immediate clinical impact, given that parallel imaging is the standard for accelerated MRI in modern scanners.*
>
> We thank the reviewer for this comment. We agree that multi-coil parallel imaging is standard in clinical practice. Our single-coil setting is a proof-of-concept to demonstrate the benefit of joint optimization for metal artifact scenarios. The framework can be extended to multi-coil by incorporating coil sensitivity information into the state representation. We have noted this as a future direction in our conclusion.
>
> ---
>
> >W3: *Focusing solely on hip implants restricts the evaluation of the policy's robustness, leaving it unclear how the method handles more complex geometries like spinal fixation or dental hardware.*
>
> We thank the reviewer for this comment. In the revised paper, we added **cross-dataset evaluation** using a **different hip implant geometry**, and MASC still generalizes well (**Table 3**). This shows our method can handle variations in implant shape rather than overfitting to one specific geometry. The same framework could be applied to other implant types like spinal or dental hardware with appropriate training data.
>
> ---
>
> >W4: *Comparing primarily against static sampling baselines (Random, Center-out) overlooks stronger recent deep learning-based reconstruction baselines (e.g., VarNet or unrolled networks) that might handle undersampling artifacts better even without active selection.*
>
> We thank the reviewer for this suggestion. We would like to clarify that we do compare with learning-based acquisition methods: **SS-DDQN** in our original submission and **DQN** added in the revision. VarNet and unrolled networks are reconstruction methods rather than acquisition strategies. In our work, all methods use the same IFFT reconstruction to ensure fair comparison. Our main contribution focuses on learning what k-space lines to acquire rather than the reconstruction method itself.
>
> ---
>
> >Detailed comments 1: *Discussing the inference latency of the PPO agent is crucial, as active acquisition requires real-time decision-making during the TR intervals of the scanner.*
>
> We thank the reviewer for this practical point. Our PPO agent is a lightweight CNN, so each forward pass is fast. However, we acknowledge that real-time integration with scanner hardware requires further engineering effort. This is a common challenge for all active acquisition methods including DQN and SS-DDQN.
>
> ---
>
> >Detailed comments 2: *Expanding the discussion on how the specific simulation parameters (e.g., readout bandwidth) influence the difficulty of the task would provide better context for the results.*
>
> We thank the reviewer for this thoughtful suggestion. In the revised paper, we added a discussion in **Appendix D** on how simulation parameters like readout bandwidth, field strength, and implant material affect artifact severity and task difficulty. Our parameter choices represent a typical and challenging clinical scenario for hip imaging at 3T.

---

> ### Author Response · Authors · 2026-01-25
> **Response to Reviewer rFC2 (2/2)**
>
> >Detailed comments 3: *Clarifying whether the reward function's dependence on SSIM and NMSE drives the agent to prioritize high-frequency edges or low-frequency contrast would help interpret the learned masks.*
>
> We thank the reviewer for this insightful question. We have added clarification in **Section 2.2** (Co-Adaptive Training). Our reward function combines SSIM and NMSE with equal weights (λ_ssim = λ_nmse = 0.5). SSIM captures structural and perceptual quality including edges, while NMSE measures overall intensity accuracy. This balanced combination encourages the agent to acquire k-space lines that benefit both aspects. As shown in **Figure 4**, the learned masks sample both central low-frequency regions and selected high-frequency lines, suggesting a balanced strategy.
>
> ---
>
> >Detailed comments 4: *Providing more details on the "invalid actions masked" step in the policy network would clarify how the agent handles the budget constraints during the rollout.*
>
> We thank the reviewer for this question. We have added clarification in **Section 2.2**. At each step, the actor outputs probabilities over all k-space lines, but we mask already-acquired lines to zero before sampling. This ensures the agent only selects from unacquired lines. The acquisition continues step by step until the budget is exhausted (e.g., 18 lines for 10× acceleration).
>
> ---
>
> >Questions to address in the rebuttal 1: *How does the framework perform when tested on implant materials or geometries significantly different from the training set (e.g., Titanium vs. CoCr)?*
>
> We thank the reviewer for this question. We added **cross-dataset evaluation** using a different hip implant geometry than training. MASC still generalizes well on this unseen implant (**Table 3**), showing our method can handle variations in implant shape. Testing on different materials like Titanium is an interesting direction to explore.
>
> ---
>
> >Questions to address in the rebuttal 2: *Is there a straightforward pathway to extend this method to multi-coil parallel imaging, and does the state space explosion become a bottleneck?*
>
> We thank the reviewer for this question. As we mentioned in W2, the framework can be extended to multi-coil by incorporating coil sensitivity information into the state representation. The action space (selecting phase-encoding lines) stays the same, so state space explosion is not a major concern. Our single-coil setting is a proof-of-concept, and multi-coil extension is noted as a future direction in our conclusion.
>
> ---
>
> >Questions to address in the rebuttal 3: *Have you attempted to apply the trained policy to any real phantom data (even without ground truth) to qualitatively assess if the simulation-learned features transfer to real k-space?*
>
> We thank the reviewer for this question. As we addressed in W1, we added **cross-dataset evaluation on FastMRI** in the revised paper. This uses real clinical MRI scans from NYU Langone Health with a different artifact simulation method than our training data. MASC generalizes well on this unseen data (**Table 3**), providing evidence that simulation-learned features can transfer.

---

> ### Comment · Area_Chair_dDid · 2026-02-01
> **For Reviewer - Please update your final rating after reviewing the author's response.**
>
> Hello there, please update your final rating after reviewing the author's response. Thank you for your time and support.

---

### Official Review · Reviewer_UfV1 · 2026-01-09

**Confidence:** 4
**Preliminary Rating:** 2

**Summary:**

This paper proposed a unified reinforcement learning framework that jointly optimizes metal-aware k-space sampling and metal artifact reduction for accelerated MRI. The experiments are done on a simulated phantom dataset and PPO is used as the RL algo. Experiments, along with ablation studies show that the proposed method outperform the baselines (random, equispace, SSDQN...).

**Strengths:**

The authors explored in the regime of using PPO for metal artifact reduction for accelerated MRI. Experimental evaluation includes quantitative metrics and qualitative comparisons, making the improvements interpretable. Also the paper is well written and easy to follow.

**Weaknesses:**

The proposed method is clinical infeasible and computing heavy.
1. All experiments are done on a simulated phantom dataset and no realistic clinical dataset is used. In real metal-implant dataset, there're more artifacts and lower SNR than phantom simulated dataset. I doubt the generalized ability of RL based network trained on simulated dataset.
2. No comparisons between supervised-based methods. Only naive traditional methods are compared.
3. Inference computing burden is clinical infeasible. It will not enable online sampling.
4. Implant type is limited.

**Detailed Comments:**

If resources allowed, I'd like to see more results on diverse implant types, comparisons between hybrid physics-informed or model-based deep learning approaches and realistic dataset.

**Justification Of The Preliminary Rating:**

The novelty is incremental and the evaluation scope somewhat limited. The proposed method is not clinical feasible.
I could still see the efforts from the authors: extensive experiments and generating simulated dataset from the phantom.

**Questions To Address In The Rebuttal:**

As stated in the comments above, I'd like to see more results from the authors about the comparisons with model-based / learning based methods and realistic dataset.

---

> ### Author Response · Authors · 2026-01-25
> **Response to Reviewer UfV1 (1/1)**
>
> We would like to thank you for recognizing the strengths of our paper, including that "experimental evaluation includes quantitative metrics and qualitative comparisons, making the improvements interpretable", "the paper is well written and easy to follow", and "I could still see the efforts from the authors: extensive experiments and generating simulated dataset from the phantom". In the following, we provide point-by-point responses to all your concerns.
>
> ---
>
> >W1: *All experiments are done on a simulated phantom dataset and no realistic clinical dataset is used. In real metal-implant dataset, there're more artifacts and lower SNR than phantom simulated dataset. I doubt the generalized ability of RL based network trained on simulated dataset.*
>
> We thank the reviewer for this important concern. To address the generalization ability, we conducted **additional cross-dataset experiments** using realistic clinical MRI data. It's a big challenge that there is no existing open-source dataset that provides paired metal-artifact and artifact-free MRI images with k-space data. This is because: (1) Real metal-artifact cases cannot have artifact-free ground truth (the implant cannot be removed) (2) k-space data is rarely preserved in clinical archives (3) Paired data would require imaging the same anatomy with and without implant, which is clinically infeasible. For these reasons, physics-based artifact simulation on real clinical data is often used in the MAR research community.
>
> To directly address the reviewer's concern, we used the **FastMRI knee dataset**, which contains realistic clinical MRI scans from NYU Langone Health (real patient data, not synthetic phantom), combined with physics-based metal artifact simulation following a published work in MICCAI 2018. The details have been added to our **Data section**. Results are shown in **Table 3** and the results section, demonstrating that our RL-based approach successfully generalizes to realistic clinical MRI data with independently simulated metal artifacts.
>
> ---
>
> >W2: *No comparisons between supervised-based methods. Only naive traditional methods are compared.*
>
> We thank the reviewer for this comment. We want to clarify that we do compare with learning-based methods, not just traditional baselines. Our original submission included **SS-DDQN**, an RL-based acquisition policy for active MRI sampling. In the revised paper, we have added **DQN** as an additional learning-based baseline.
>
> We also now evaluate all baselines **with and without MAR post-processing** (**Table 1**). Adding MAR essentially creates two-stage supervised pipelines. MASC still outperforms the best two-stage baseline.
>
> ---
>
> >W3: *Inference computing burden is clinical infeasible. It will not enable online sampling.*
>
> We thank the reviewer for this practical concern. We agree that our current implementation is a proof-of-concept rather than a clinical-ready system. That said, the inference only involves forward passes through a lightweight CNN policy and U-Net, which is faster than many iterative reconstruction methods.
>
> It's also worth noting that this sequential decision-making overhead is common to all active acquisition methods, including DQN and SS-DDQN. Improving inference speed through techniques like network pruning or selecting multiple lines at once is an interesting direction for future work.
>
> ---
>
> >W4: *Implant type is limited.*
>
> We thank the reviewer for this comment. In the revised paper, we added **cross-dataset evaluation** using a **different hip implant geometry** than training, which produces different artifact patterns. MASC still outperforms all baselines on this unseen implant (**Table 3**), showing that our method can generalize across implant variations. Extending to more implant types and anatomical regions is an interesting direction for future work.
>
> ---
>
> >Detailed comments: *If resources allowed, I'd like to see more results on diverse implant types, comparisons between hybrid physics-informed or model-based deep learning approaches and realistic dataset.*
>
> We thank the reviewer for this constructive suggestion. As stated above, in the revised paper we added **DQN** as an additional learning-based baseline alongside our original SS-DDQN comparison. We also added **cross-dataset evaluation** using a different hip implant geometry, showing that MASC generalizes to unseen implant variations (**Table 3**). The **FastMRI-based Metal Artifacts Dataset** uses real clinical MRI scans with simulated artifacts, which is a step toward more realistic data.

---

> ### Comment · Area_Chair_dDid · 2026-02-01
> **For Reviewer - Please update your final rating after reviewing the author's response.**
>
> Hello there, please update your final rating after reviewing the author's response. Thank you for your time and support.

---

### Official Review · Reviewer_kHKs · 2026-01-13

**Confidence:** 3
**Preliminary Rating:** 3
**Final Rating:** 4

**Summary:**

The paper proposes MASC, a network that jointly addresses *metal artifact reduction (MAR)* and *MRI acceleration* (through sparse k-space acquisition). The approach combines a U-Net-based MAR network with a proximal policy optimization (PPO)-based reinforcement learning (RL) agent that learns to decide which k-space data to sample next. Both networks are jointly trained on a paired dataset of simulated k-space data with and without virtual metal implants. The authors compare their method to simple k-space sampling strategies, as well as a learning based approach (all without MAR).

**Strengths:**

The paper is generally easy-to-follow, and addresses a relevant problem. The proposed method is well-explained and supported by helpful figures. The main strength of this paper are:
- **Clever combination of existing approaches**: Exploring the combination of MAR and RL-based k-space sampling is interesting. The idea of jointly training both networks is simple, but seems to actually be quite effective (as demonstrated in the ablation study). While the evaluation is limited in scope (covering only one anatomical region and implant type), the results motivate further exploration of this approach.
- **Open resources**: The model is trained on a publicly available dataset, and the authors claim to publish their code, which should make the proposed method reproducible and easily applicable.

**Weaknesses:**

While the paper is generally interesting, it also has several weaknesses:
- **Weak evaluation/ comparing methods**: The authors only compare their method to very simple k-space sampling strategies (random, random with low-frequency bias, equidistant, center-out) and only one learning-based approach (SS-DDQN). Additionally, these MRI acceleration strategies are applied without, but compared to a method with MAR which seems to be an unfair comparison. A comparison to even naive combinations of sampling strategy + accelerated acquisition would make it much easier to judge the actual performance.
- **Some questionable choices that should be explained further**: The authors state that they pretrain their MAR network using L1 and SSIM (however without saying how these two components are weighted). Later they state that during the joint fine-tuning, they only use L2. I don't understand this choice and can't find a rationale for it. Even later they also state that the MAR network is pretrained with the loss described in equitation 6. This equation however does not exist. What is the actual training setup?
- **Missing information on the network architecture**: Adding some information on the network architectures beyond only stating that e.g. a residual U-Net is used would be really helpful.
- **A lot of minor mistakes**: Even though the paper is easy-to-follow, there is quite some minor mistakes (that I also address in the "Detailed Comments" section) including notation inconsistencies, referring to equations that don't exist or sentences with doubled words. It seems like the paper was written in a hurry and the authors should make sure to correct these careless mistakes.

**Detailed Comments:**

There is some more comments/ suggestions:
- The notation between the main paper and the algorithm in the appendix is not consistent, e.g. mask is describes as $M$ or $m$, images in the main paper are indexed with $t, t+1, ...$. In the appendix the authors also use $prev, recon$ for indexing. Make sure the notation is consistent.
- Some abbreviations are never introduced, e.g. GAE, DQN. Make sure to consistently do this.
- Some sentences are a bit weird. In Section 2.2 for example you talk about an "artifact-aware PPO-based artifact-aware acquisition policy". Make sure to correct these mistakes.
- I'd additionally be interested in inference times. As your MAR-network is basically part of the acquisition pipeline this is relevant, especially when the paper addresses accelerated MRI.
- Experimenting with different acquisition budgets could also be an interesting extension of the work.

**Justification Of Final Rating:**

The rebuttal addressed some of my concerns and the additional experiments strengthen the paper. Some limitations remain, but I think this paper demonstrates the advantages of jointly training MAR and RL policy. I agree with the other reviewer that extending this work to a multi-coil setting would be needed to make a real clinical impact. However, I'm willing to raise my score and would like to thank the authors for their detailed comments.

**Justification Of The Preliminary Rating:**

While the general idea and the proposed approach are an interesting combination of methods, the lack of a fair comparison makes it hard to judge the actual performance of the method. Additional weaknesses, including missing details on the network architecture and an unclear pretraining/fine-tuning loss for the MAR network, make this a borderline paper.

**Questions To Address In The Rebuttal:**

I'd like the authors to focus their rebuttal on the main weaknesses of the paper. Additionally the minor mistakes stated in the "Detailed Comments" section should be addressed in a final version of this paper.

---

> ### Author Response · Authors · 2026-01-25
> **Response to Reviewer kHKs (1/2)**
>
> We would like to thank you for recognizing the strengths of our paper, including that “The paper is generally easy-to-follow, and addresses a relevant problem", "The proposed method is well-explained and supported by helpful figures.", “Open resources”. In the following, we provide point-by-point responses to all your concerns.
>
> ---
> >W1: *Weak evaluation/ comparing methods: The authors only compare their method to very simple k-space sampling strategies (random, random with low-frequency bias, equidistant, center-out) and only one learning-based approach (SS-DDQN). Additionally, these MRI acceleration strategies are applied without, but compared to a method with MAR which seems to be an unfair comparison. A comparison to even naive combinations of sampling strategy + accelerated acquisition would make it much easier to judge the actual performance.*
>
> We thank the reviewer for pointing out the insufficient evaluation process. For the **baseline methods**, we add **one more widely-used learning-based baseline** (**Table 1** in paper): RL with DQN policy in addition to the original standard k-space sampling strategies (random, low-frequency bias, equidistant, center-out) and the learning-based approach SS-DDQN.
>
> For the fair comparison with MAR, we agree this is important. We have added **additional experiments** (full table, also has been added to the results section in paper) where **all baseline sampling strategies are combined with MAR post-processing** (extended **Table 1**). Results show our joint optimization still outperforms these naive ”sampling + MAR” combinations. It shows that the gain from co-training is significant to artifact reduction, rather than simply adding MAR as a post-processing step.
>
> >W2: *Some questionable choices that should be explained further: The authors state that they pretrain their MAR network using L1 and SSIM (however without saying how these two components are weighted). Later they state that during the joint fine-tuning, they only use L2. I don't understand this choice and can't find a rationale for it. Even later they also state that the MAR network is pretrained with the loss described in equitation 6. This equation however does not exist. What is the actual training setup?*
>
> We apologize for the unclear statement and wrong equation reference. We have **added more explanation** in **Section 2.2** about the loss choices. During pretraining, the combination of L1 and SSIM is effective for training from scratch: L1 provides robust supervision against outliers while SSIM captures perceptual quality. During finetuning, MSE is chosen because it provides more stable gradients than L1, which helps stabilize training when combined with noisy policy gradient updates. At this stage, the MAR network is already well-initialized and only needs mild regularization. The equation reference in **Section 3.2** has also been corrected.
>
> ---
>
> >W3: *Missing information on the network architecture: Adding some information on the network architectures beyond only stating that e.g. a residual U-Net is used would be really helpful.*
>
> We thank the reviewer for raising this point. We have **added more description** in **Section 2.2**. The MAR network is a U-Net with four encoder and four decoder stages. Each encoder stage doubles the feature channels (64, doubling at each stage to 1024) using two 3×3 convolutions with batch normalization and ReLU, followed by max pooling. The decoder uses bilinear upsampling with skip connections. The network employs residual learning where $g_\psi(I) = I + r_\psi(I)$, allowing it to focus on learning artifact corrections rather than full image reconstruction.
>
> ---
>
> >W4 & Detailed comments 1-3: *A lot of minor mistakes: Even though the paper is easy-to-follow, there is quite some minor mistakes including notation inconsistencies, referring to equations that don't exist or sentences with doubled words. The notation between the main paper and the algorithm in the appendix is not consistent. Some abbreviations are never introduced, e.g. GAE, DQN. Some sentences are a bit weird, e.g. "artifact-aware PPO-based artifact-aware acquisition policy".*
>
> We appreciate the reviewer's constructive suggestions. We have **unified the notation** throughout the paper and appendix - the mask is now consistently denoted as $M$, and indexing is consistent between the method section and appendix. We have ensured that **all abbreviations** (GAE, DQN) are properly introduced upon first use. We have also **revised redundant wording** and other writing issues.

---

> ### Author Response · Authors · 2026-01-25
> **Response to Reviewer kHKs (2/2)**
>
> >Detailed comments 4: *I'd additionally be interested in inference times. As your MAR-network is basically part of the acquisition pipeline this is relevant, especially when the paper addresses accelerated MRI.*
>
> We appreciate the reviewer's comment on inference time. Our pipeline involves forward passes through a lightweight CNN policy and U-Net, which is relatively fast (~0.1s per slice). However, we acknowledge that real-time integration with scanner hardware requires further engineering effort. This sequential decision-making overhead is common to all active acquisition methods including DQN and SS-DDQN. Our current work focuses on demonstrating the benefit of joint optimization, and real-time implementation is an important next step toward clinical deployment.
>
> ---
>
> >Detailed comments 5: *Experimenting with different acquisition budgets could also be an interesting extension of the work.*
>
> We appreciate the reviewer's suggestion. We have **added experiments** with **5× acceleration** (8 initial lines + 32 budget), which is quite different from our original 10× setting. Results are shown in **Appendix C Table 5**, demonstrating that MASC consistently outperforms baselines across varying acceleration factors. We also observe that the performance gap between MASC and baselines narrows at lower acceleration, as the increased k-space budget reduces the importance of optimal line selection.

---

> ### Comment · Area_Chair_dDid · 2026-02-01
> **For Reviewer - Please update your final rating after reviewing the author's response.**
>
> Hello there, please update your final rating after reviewing the author's response. Thank you for your time and support.

---

### Author Rebuttal · Authors · 2026-01-25

**Rebuttal:**

## Rebuttal Summary

Dear Reviewers and Area Chairs,

We thank all reviewers for their constructive feedback. We have made substantial revisions including: (1) cross-dataset evaluation on FastMRI with a different implant geometry, (2) DQN as an additional learning-based baseline, and (3) comparison with and without MAR post-processing. All new content is highlighted in orange.

---

### Major Additions

* **Cross-dataset generalization:** We evaluated on the **FastMRI-based Metal Artifacts Dataset** with real clinical MRI scans and a different hip implant geometry. MASC maintains superior performance on this unseen data. See `new Table 3, Section 4.3`.

* **Expanded baselines:** We added **DQN** and evaluated all baselines **with and without MAR**, enabling fair comparison against two-stage pipelines. See `revised Table 1, Section 4.1`.

* **5× acceleration experiments:** Demonstrates consistent improvements across varying acquisition budgets. See `new Appendix C, Table 5`.

* **Simulation parameter discussion:** Analysis on how readout bandwidth, field strength, and implant material affect task difficulty. See `new Appendix D`.

---

### Summary Tables

**New Content**

| Addition | Location | Addresses |
|----------|----------|-----------|
| Cross-dataset evaluation on FastMRI | Table 3, Section 4.3, Section 3.1 | R2-W1, R3-W1/W3 |
| 5× acceleration experiments | Appendix C, Table 5 | R1-Detailed 5 |
| Simulation parameter discussion | Appendix D | R3-Detailed 2 |

**Revised Content**

| Revision | Location | Addresses |
|----------|----------|-----------|
| DQN baseline + MAR comparison | Table 1, Section 4.1 | R1-W1, R2-W2, R3-W4 |
| Loss function, architecture, masking | Section 2.2 | R1-W2/W3, R3-Detailed 4 |
| Notation & abbreviations | Throughout | R1-W4/Detailed 1-3 |


---

We believe our revisions have addressed all major concerns.

Best Regards,

The MASC Team

**Supporting Material:**

/attachment/ec0eb05b9a0277e64b9ff29333bdac31392e0292.pdf

---

### Meta-Review · Area_Chair_dDid · 2026-02-09

**Recommendation:** Accept (Oral)
**Confidence:** 5

**Metareview:**

This paper presents MASC, a unified reinforcement learning framework that jointly optimizes metal-aware k-space sampling and metal artifact reduction for accelerated MRI. The work addresses an important and underexplored problem at the intersection of active MRI acquisition and metal artifact reduction, which are typically treated in isolation. The proposed formulation is technically sound, well motivated, and clearly presented.

Across reviewers, there is strong agreement that the joint optimization of the sampling policy and MAR network is the key contribution, and that the physics-based paired simulation framework is a necessary and well-justified solution to the lack of real ground-truth data in metal-implant MRI. The ablation studies convincingly demonstrate that end-to-end co-adaptive training yields consistent and non-trivial gains over both conventional sampling strategies and naïve combinations of sampling with post-hoc MAR. Visualizations and acquisition trajectories further support the physical plausibility of the learned policies.

While initial reviews raised concerns regarding evaluation scope, fairness of comparisons, architectural clarity, and reliance on simulated data, these issues were substantially addressed in the rebuttal and revised manuscript. The authors added stronger learning-based baselines, included fair comparisons where all sampling strategies are paired with MAR, clarified loss functions and network architecture, and corrected numerous presentation issues. Importantly, the added cross-dataset evaluation on FastMRI-based clinical data with independently simulated artifacts alleviates major concerns about overfitting to a single simulator and strengthens the generalization claims.

---

### Decision · Program_Chairs · 2026-02-13

Accept (Poster)